# Sudden Unexpected Postnatal Collapse and Therapeutic Hypothermia: What’s Going On?

**DOI:** 10.3390/children9121925

**Published:** 2022-12-08

**Authors:** Luca Bedetti, Licia Lugli, Elisabetta Garetti, Isotta Guidotti, Maria Federica Roversi, Elisa Della Casa, Francesca Miselli, Maria Carolina Bariola, Antonella Di Caprio, Marisa Pugliese, Fabrizio Ferrari, Alberto Berardi

**Affiliations:** 1PhD Program in Clinical and Experimental Medicine, University of Modena and Reggio Emilia, 41121 Modena, Italy; 2Neonatal Intensive Care Unit, University Hospital of Modena, 41125 Modena, Italy; 3Pediatric Post-Graduate School, University Hospital of Modena and Reggio Emilia, 41125 Modena, Italy; 4Psychology Unit, University Hospital of Modena, 41125 Modena, Italy

**Keywords:** sudden unexpected postnatal collapse, therapeutic hypothermia, brain cooling, neurodevelopmental outcome, cerebral palsy

## Abstract

Sudden unexpected postnatal collapse (SUPC) is a rare event, potentially associated with catastrophic consequences. Since the beginning of the 2000s, therapeutic hypothermia (TH) has been proposed as a treatment for asphyxiated neonates after SUPC. However, only a few studies have reported the outcome of SUPC after TH. The current study presents the long-term neurodevelopmental outcome of four cases of SUPC treated with TH in a single Italian center. Furthermore, we reviewed the previous literature concerning 49 cases of SUPC treated with TH. Among 53 total cases (of whom four occurred in our center), 15 (28.3%) died before discharge from the NICU. A neurodevelopmental follow-up was available only for 21 (55.3%) out of the 38 surviving cases, and seven infants developed neurodevelopmental sequelae. TH should be considered in neonates with asphyxia after SUPC. However, SUPC is a rare event, and there is a lack of comparative clinical data to establish the risk/benefit of TH after SUPC with different degrees of asphyxia. Analysis of large cohorts of newborns with SUPC, whether treated with TH or untreated, are needed in order to better identify infants who should undergo TH.

## 1. Introduction

Sudden unexpected postnatal collapse (SUPC) is a rare event, potentially associated with catastrophic consequences [1]. SUPC is usually defined as the event occurring in term or near term (>35 weeks’ gestation) healthy infants, who unexpectedly collapse within the first seven days of life, with need for resuscitation, intermittent positive pressure ventilation, subsequent requirement of intensive care, and further potential development of encephalopathy or death [1].

The incidence of SUPC varies widely, from 2.6 to 34/100,000 live births, depending on the different definitions (e.g., type of resuscitation required or timing of occurrence of the event) and the type of the study (multiple- or single-center studies) [2]. The actual incidence is likely higher, given the probable underreporting of less critical cases [3], and it does not seem to have been changed by the back to sleep campaign of the mid 1990s [4]. Because SUPC has been described as more frequent following the widespread use of skin-to-skin practice [5], guidelines for its prevention have been developed [6,7], and interest has increased regarding the therapeutic options for affected neonates. Particular attention has been paid to therapeutic hypothermia (TH), which is used in hypoxic–ischemic encephalopathy (HIE) [8,9]. Indeed, the brain damage associated with SUPC seems similar to that of HIE, with comparable clinical and neuroradiological findings [10]. Unfortunately, the advantages of TH after SUPC have not been demonstrated through randomized clinical trials, and data regarding neonates undergoing TH after SUPC or their long-term neurodevelopmental outcome are limited.

We reviewed, in detail, cases of SUPC undergoing TH who were admitted to our Neonatal Intensive Care Unit (NICU) during a 4-year period. Furthermore, we reviewed cases of SUPC treated with TH which had been reported in the literature.

## 2. Materials and Methods

The NICU of Azienda Ospedaliero Universitaria di Modena (Modena, Italy) is a level 3, high volume center where critical newborns from the entire district are referred. The district of Modena consists of five centers (level 3, *n* = 1; level 2, *n* = 2; level 1, *n* = 2), with approximately 6000 live births/year.

This is a retrospective analysis of cases of SUPC undergoing TH from 1 December 2014 to 1 December 2019. Furthermore, we reviewed the scientific literature to define the evidence supporting TH after SUPC and to describe, in detail, the outcome of treated infants reported by other studies.

## 3. Results

### 3.1. SUPC Cases from 2014 to 2019

During the study period, 27,450 live births were delivered in the district of Modena. From 2014 to 2019, six infants were admitted to NICU because of SUPC (estimated incidence of SUPC, 21.8 per 100,000 live births), of which four underwent TH; the remaining two infants had a normal polygraphic-EEG (p-EEG) with mild (*n* = 1) or no HIE (*n* = 1), and were not treated.

All four infants were born full-term, were Caucasian, and had an Apgar score of 10 at the 10th minute. Table 1 shows the maternal and neonatal data of infants who underwent TH.

SUPCs occurred at a median age of 95 min, and all infants had been positioned prone for skin-to-skin care. After SUPC, they required cardiopulmonary resuscitation with orotracheal intubation (Table 2). Three cases occurred when the newborn was still in the delivery room with their mother, but one case occurred during rooming-in, after the newborn had been sent to the obstetric unit with her mother. All had severe metabolic acidosis and underwent repeated neurological examinations and prolonged p-EEG recording. HIEs ranged from 1 to 3, and their p-EEGs showed moderate to severe abnormalities. TH was started within 6 h of SUPC. Continuous p-EEG recording was performed during TH: in case 1, p-EEG abnormalities remained severe during the entire period of TH (Figure 1). In the remaining cases, p-EEGs were moderate-to-severe on admission, but background p-EEGs improved during TH without any evidence of clinical or electrical seizures (Figure 2). TH was discontinued after 72 h without adverse effects. The duration of mechanical ventilation ranged from 2 to 7 days.

All SUPC were unexplained, since infectious, metabolic, and cardiac underlying diseases were ruled out. The hospital stays ranged from 19 to 59 days. All parents were trained in basic pediatric resuscitation, and all infants were discharged with home monitoring of vital functions.

Table 2 shows clinical findings, brain MRI findings, and neurodevelopmental outcomes of the described cases. According to the neuroimaging protocol in use in our NICU for perinatal asphyxia, each patient underwent a brain MRI in the acute phase (<8 days of life) and a subsequent control from 28 days of life onwards. In addition, case 1 underwent a follow-up MRI at 30 months of age. TheMRI was performed using a Philips Intera 1.5-T MRI scanner (Philips Medical Systems, Best, The Netherlands). MRI findings ranged from normal (cases 2 and 3, Figure 3) to mild focal lesions (case 4, Figure 4) and severe lesions in the basal ganglia and thalami (case 1, Figure 5). At the age of 24 months, case 1 had spastic cerebral palsy, with post-neonatal epilepsy and severe cognitive delay, while the remaining three infants were within a normal range.

### 3.2. Literature Review

So far, 53 infants (including our four cases) have been treated with TH for SUPC [2,5,12,13,14,15,16,17,18]. The 49 cases already reported in the literature occurred during the period between 2006 and 2018. The definitions of SUPC between the studies could differ in terms of timing of the event, because some limited the presentation of SUPC to the first 24 h of life or less [5,12,14,15]. Nevertheless, all infants described met the definition criteria by Becher et al. [1]. In total, 15 infants died, 17 survived without follow-up information, and only 21 infants (including the four cases here reported) had neurodevelopmental follow-up data (Table 3). Only 12 of 21 underwent cerebral MRI.

Neurodevelopmental outcome was within a normal range in 14 of 21 infants, and impaired (with a wide range of severity) in the remaining seven infants (of which two developed cerebral palsy, two developed moderate neurodevelopmental delay, and three had an unspecified “poor outcome”).

## 4. Discussion

TH has been proven to reduce death, severe cerebral lesions, and disabilities in asphyxiated full-term infants [19]; however, its use in infants with SUPC is infrequently reported, and its long-term neurodevelopmental outcome is not well known.

We report four cases of infants affected by SUPC who were treated with TH, as well as their long-term neurodevelopmental outcomes. All infants required resuscitation, but the severity of their asphyxia was variable. TH was performed according to Italian guidelines for TH in HIE [20], which suggest TH after SUPC according to the severity of asphyxia.

To date, the rarity of SUPC has precluded randomized clinical trials of TH. Furthermore, it may be unethical not to treat, since TH has proven to be beneficial in perinatal asphyxia. Indeed, brain lesions after SUPC are comparable to those related to perinatal HIE. After SUPC, severe EEG abnormalities and severe injuries to the central grey matter and brainstem have been reported [10]. In addition, unlike perinatal HIE, where the timing of the insult is sometimes uncertain, the timing of SUPC is known, giving an opportunity to perform TH within the recommended 6-hour window of the asphyxia [21].

Concerning the 53 total cases (including our four cases) treated with TH and so far described in the literature [2,5,12,13,14,15,16,17,18], neurodevelopmental follow-up data were available only for 55% of survivors, whereas the severity of asphyxia and cerebral MRI findings were not always reported. Rates of both mortality and neurodevelopmental impairment among SUPC survivors undergoing TH were high (28% and 33%, respectively), suggesting that the outcome of SUPC could be poorly affected by TH. However, this conclusion could be biased, as only the most severe cases may have been reported. Instead, it is probable that the efficacy of TH after SUPC is at least equal to or greater than that concerning cases of perinatal encephalopathy. In fact, in some of the latter, the timing of the insult is not always known, and could have begun beyond the 6-hour time window usually recommended. In contrast, in infants who are asphyxiated after SUPC, the timing of the insult is generally well known and very close. Notably, among our four cases of SUPC, three had had moderate-to-severe abnormalities at p-EEG recording on admission, but p-EEG normalized during TH and neurodevelopmental outcome was normal, highlighting the potential beneficial effects of TH.

Since the incidence of SUPC is estimated to be from 2.6 to 34/100,000 live births, and because approximately 3,780,000 infants were born in Europe in 2019 [22], the number of cases undergoing TH currently reported in the literature seems largely underestimated, and follow-up data from large cohorts are welcome [18].

## 5. Conclusions

SUPC may lead to asphyxia of varying severity that may require TH. Only large prospective observational studies will more accurately describe the current clinical practice and the long-term outcomes after TH.

## Figures and Tables

**Figure 1 children-09-01925-f001:**
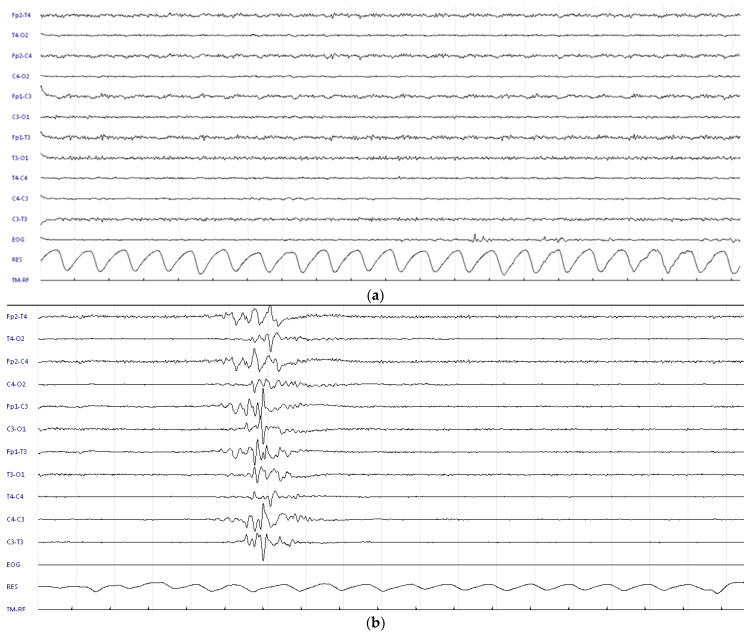
p-EEG in case 1 (**a**) at recruitment: severe p-EEG depression with superimposed muscular activity; (**b**) at 24 h during TH: burst suppression pattern with long inter-burst periods.

**Figure 2 children-09-01925-f002:**
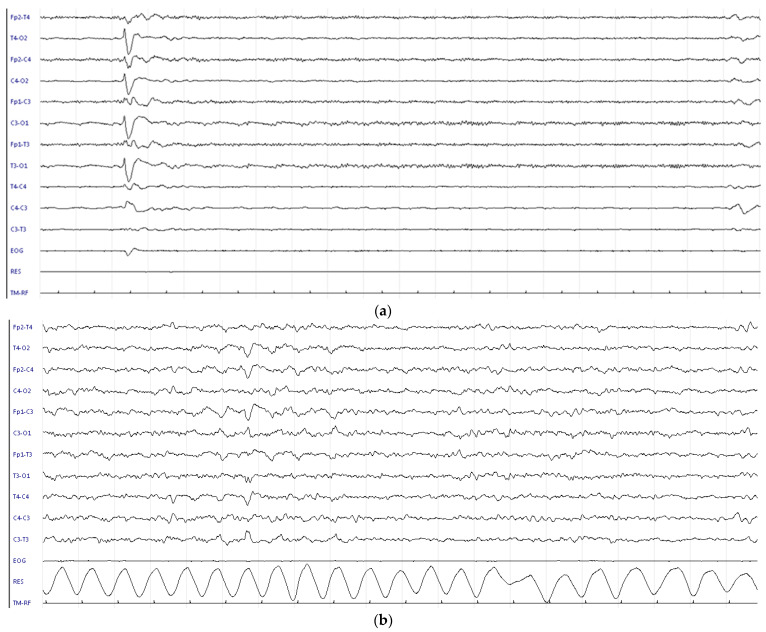
p-EEG in case 2 (**a**) at recruitment: severe p-EEG depression with superimposed muscular activity; (**b**) at 24 h during TH: mild to normal pEEG.

**Figure 3 children-09-01925-f003:**
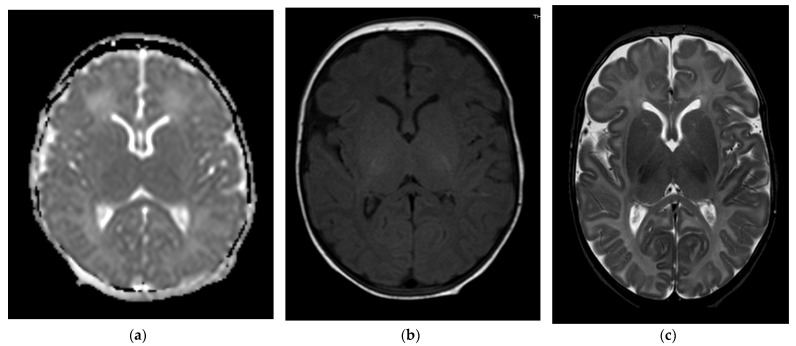
Brain MRI scans of case 2. (**a**) At 5 days of life, DWI showed a normal apparent diffusion coefficient of the brain. (**b**,**c**) At 40 days of life, T1- and T2-weighted axial spin echo sequence shows normal signal intensity of the brain.

**Figure 4 children-09-01925-f004:**
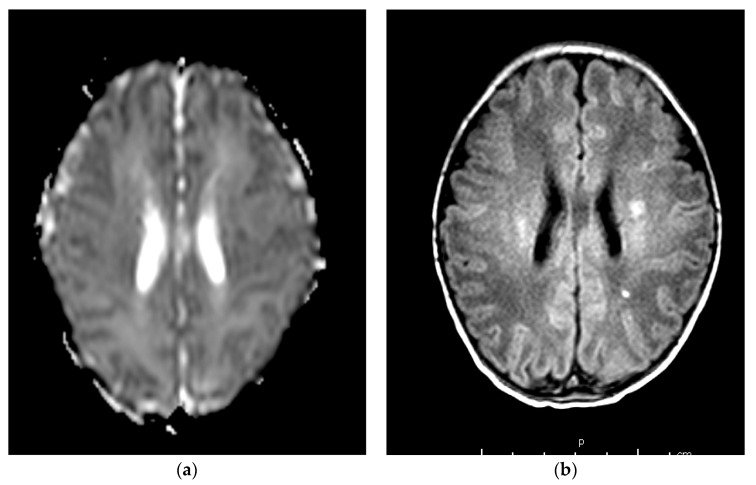
Brain MRI scans of case 4. (**a–c**) At 5 days of life, T1-weighted axial spin echo sequence (**b**) showed small foci of increased signal intensity in the fronto-parietal-occipital white matter (arrows), with a reduced apparent diffusion coefficient on DWI (arrows, **a**). Gradient echo sequence (**c**) shows a prominent aspect of the profound medullary veins. (**d**) At 64 days of life, T2-weighted and SWI sequences show reduced extension and appearance of the focal white matter lesions, with a slight increase in the ventricular system.

**Figure 5 children-09-01925-f005:**
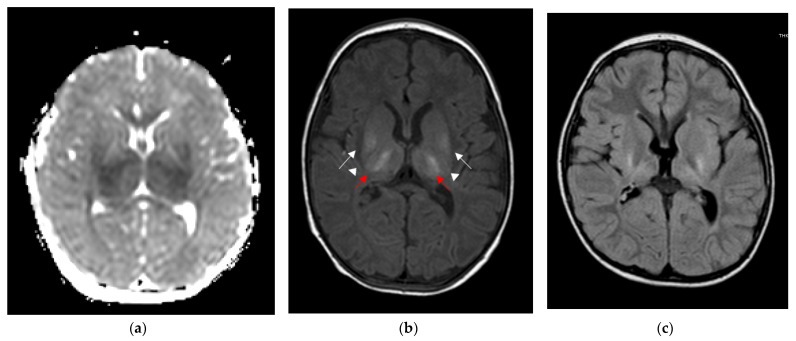
Brain MRI scans of case 1. (**a**) At 5 days of life, DWI showed a symmetric reduction in the apparent diffusion coefficient of thalami and putamen. (**b**) At 21 days of life, T1-weighted axial spin echo sequence showed severe bilateral abnormally increased signal intensities in the ventro-lateral thalamic nuclei (red arrows) and posterior aspect of the putamen (white arrow), with loss of normal signal intensity in the posterior limb of the internal capsule (arrowheads). (**c**) FLAIR sequence at 30 months showed atrophic and gliotic evolution of thalami and lenticular nuclei, and quantitative reduction in the posterior periventricular white matter.

**Table 1 children-09-01925-t001:** Maternal and neonatal data of infants with SUPC who underwent TH.

Cases	Pregnancy Course	Maternal Age, Years	Drugs Administered during Pregnancy	Mode of Delivery	Outborn	Sex	Gestational Age, Weeks + Days	Birth Weight, g
1	Gestational diabetes, obesity, hyperthyroidism	39	Methimazole	VD	Yes	male	40 + 6	3250
2	Obesity	34	No	VD	Yes	male	41 + 6	3500
3	No	42	No	VD	No	male	40 + 6	2970
4	No	34	No	ECS	No	male	37 + 3	2740

ECS, emergency caesarean section; VD, vaginal delivery.

**Table 2 children-09-01925-t002:** Clinical findings and 24-month follow-up data of infants who underwent TH.

Case	Age at SUPC, Minutes	Resuscitation	pH on Admission (BE)	Lactate on Admission, mmol/L	HIE Grade *	p-EEG	MV, Days	Hospital Stay, Days	Age at First MRI, Days	Age at Second MRI, Days	MRI Findings	Outcome	GMDS Global Quotient
1	80	CPR + OT	NR	NR	3	Severely abnormal	7	59	5	28	Severe lesions of basal ganglia and thalami	Spastic cerebral palsy	NA
2	35	CPR + OT	6.8 (−24)	NR	1	Moderate/severe abnormalities	4	22	5	40	Normal	Normal	95
3	110	CPR + OT	7.2 (−13)	10.4	2	Moderate/severe abnormalities	2	22	5	45	Normal	Normal	104
4 ^§^	140	CPR + OT	6.9 (−14)	6.8	2	Moderate abnormalities	4	19	5	64	Mild focal white matter lesions	Normal ^¥^	112

BE, base excess; CPR, cardiopulmonary resuscitation; GMDS, Griffiths Mental Developmental Scales; MRI, Magnetic Resonance Imaging; MV, mechanical ventilation; NA, not applicable; NR, not reported; OT, orotracheal intubation; TH, therapeutic hypothermia. * HIE grade at admission to NICU. The severity of HIE was assessed according to the modified Sarnat and Sarnat criteria [11]. ^§^ First ECG Holter monitor found an isolated junctional pulse, which was not confirmed when the ECG Holter monitor was repeated. ^¥^ The infant underwent physiotherapy at 15 months of life due to transient motor delay.

**Table 3 children-09-01925-t003:** Characteristics of infants with SUPC treated with TH and with complete neurodevelopmental follow-up, described in the literature.

Cases	HIE Grade	Age at Follow-Up (Months)	Cerebral MRI Findings	Neurodevelopmental Outcome (Cognitive Score)
Case 1 [12]	2	12	normal	normal (NR)
Case 2 [12]	2	12	normal	normal (NR)
Case 3 [12]	2	24	normal	mild spastic cerebral palsy (NR)
Case 4 [12]	2	10	normal	normal (NR)
Case 5 [13]	3	36	normal	normal (110) ^¶^
Case 6 [15]	2	10	normal (9 months)	normal (NR)
Case 7 [18]	3	12	NR	normal (102) ^χ^
Case 8 [2]	NR	NR	small area of vascular suffering	moderate delay (NR)
Case 9 [2]	NR	NR	moderate restriction of cortico-subcortical parenchymal diffusion	moderate delay (NR)
Case 10 [17]	NR	18–20	NR	poor outcome (NR)
Case 11 [17]	NR	18–20	NR	poor outcome (NR)
Case 12 [17]	NR	18–20	NR	poor outcome (NR)
Case 13 [17]	NR	18–20	NR	normal (NR)
Case 14 [17]	NR	18–20	NR	normal (NR)
Case 15 [17]	NR	18–20	NR	normal (NR)
Case 16 [17]	NR	18–20	NR	normal (NR)
Case 17 [17]	NR	18–20	NR	normal (NR)

NR, not reported; ^¶^ Bayley scoring; ^χ^ Griffiths Mental Developmental Scales.

## Data Availability

Not applicable.

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
