# Peer review of "Sudden Unexpected Postnatal Collapse and Therapeutic Hypothermia: What’s Going On?"

_children, 2022, doi:10.3390/children9121925_

Round 1
Reviewer 1 Report
1- SUPC is a rare event - Only six have been at your center of which 4 received TH --> Please define if similar definition has been used in the prior 49 cases in literature , also what is the timeline of those 49 cases , it is not very clear
2- Line 69: clarify the incidence of SUPC per 100000 live birth
3- Authors are using HIE consistently - which is a MRI/radiological diagnosis, use of neonatal encephalopathy is suggested
4-Please mention what scoring has been utilized to grade the encephalopathy
5- What follow up testing were done , please share the bailey/developmental test scores instead of these gross terms of moderate delay and poor outcomes
Reviewer 2 Report
The authors report 4 cases with postnatal collapse who underwent hypothermia and searched the literature for other cases. This is of interest. It would be even better when they could expand the outcome data. Maybe they could JL Bass J Pediatr 2018 to the references.
Introduction, line 44, what do they mean by ‘diffusion’? maybe ‘widespread use’?
Line 60, ‘live births’ instead of ‘live birth’
Results: Line 70, what does p stands for in p-EEG?
Please add to table 1, when the collapse occurred in minutes after birth
Line 148, cognitive is now preferred rather than mental
Table 2, do they maybe have lactate data obtained on admission?
The MRIs were performed quite late, any reason why? It would be interesting to show the T1 weighted image at the level of the PLIC
Fig 4, please mention as well, that the extracerebral space is a bit enlarged.
Line 172 ‘17 survived without follow-up information’. Did they get in touch with the first author to see whether any follow-up data could be provided?
Author Response
See attached file. All changes in the manuscript are highlighted in yellow.

Round 2
Reviewer 1 Report
Suggested changes have been made